# Rapid genomic surveillance of SARS-CoV-2 in a dense urban community of Kathmandu Valley using sewage samples

Rajindra Napit[1,2], Prajwol Manandhar[1,2], Ashok Chaudhary[1], Bishwo Shrestha[1], Ajit Poudel[1,2], Roji Raut[1], Saman Pradhan[1,2], Samita Raut[1], Pragun G. Rajbhandari[1], Anupama Gurung[1], Rajesh M. Rajbhandari[1,2], Sameer M. Dixit[1], Jessica S. Schwind[3], Christine K. Johnson[4], Jonna K. Mazet[4], Dibesh B. Karmacharya[1,2,5]*

1 One Health Research Division, Center for Molecular Dynamics Nepal, Thapathali-11, Kathmandu, Nepal, 2 Virology Division, BIOVAC Nepal Pvt. Ltd., Nala, Banepa, Nepal, 3 Institute for Health Logistics & Analytics, Georgia Southern University, Statesboro, GA, United States of America, 4 One Health Institute, School of Veterinary Medicine, University of California, Davis, Davis, CA, United States of America, 5 The School of Biological Sciences, University of Queensland, Brisbane, Australia

* dibesh@cmdn.org, dibesh@biovacnepal.com

**Data Availability Statement:** All the sequence data are available in NCBI GenBank with accession number OM772935 to OM772942. All the data were not accepted to NCBI due to low genome

## Abstract

Understanding disease burden and transmission dynamics in resource-limited, low-income countries like Nepal are often challenging due to inadequate surveillance systems. These issues are exacerbated by limited access to diagnostic and research facilities throughout the country. Nepal has one of the highest COVID-19 case rates (915 cases per 100,000 people) in South Asia, with densely-populated Kathmandu experiencing the highest number of cases. Swiftly identifying case clusters (hotspots) and introducing effective intervention programs is crucial to mounting an effective containment strategy. The rapid identification of circulating SARS-CoV-2 variants can also provide important information on viral evolution and epidemiology. Genomic-based environmental surveillance can help in the early detection of outbreaks before clinical cases are recognized and identify viral micro-diversity that can be used for designing real-time risk-based interventions. This research aimed to develop a genomic-based environmental surveillance system by detecting and characterizing SARS-CoV-2 in sewage samples of Kathmandu using portable next-generation DNA sequencing devices. Out of 22 sites in the Kathmandu Valley from June to August 2020, sewage samples from 16 (80%) sites had detectable SARS-CoV-2. A heatmap was created to visualize the presence of SARS-CoV-2 infection in the community based on viral load intensity and corresponding geospatial data. Further, 47 mutations were observed in the SARS-CoV-2 genome. Some detected mutations (n = 9, 22%) were novel at the time of data analysis and yet to be reported in the global database, with one indicating a frameshift deletion in the spike gene. SNP analysis revealed possibility of assessing circulating major/minor variant diversity on environmental samples based on key mutations. Our study demonstrated the feasibility of rapidly obtaining vital information on community transmission and disease dynamics of SARS-CoV-2 using genomic-based environmental surveillance.

coverage (>50% n) those data are available in Github repository (https://github.com/INPLlab/sarscov2-wgs-sewage).

**Funding:** University of California-Davis and the USAID-funded PREDICT project for providing us with laboratory resources. Whole-genome sequencing was done at the Intrepid Nepal Genomic Center. Some of our work was partially funded by the Australian Development Agency and PSI grant (the Netherlands). Composite sampling robots were provided by Professor Eric Alm and Dr. Noriko Endo of BIOBOT (USA).

**Competing interests:** The authors have declared that no competing interests exist.

## Introduction

In the past twenty years, several diseases caused by coronavirus have posed significant global health challenges- including Severe Acute Respiratory Syndrome (SARS, 2002), Middle East Respiratory Syndrome (MERS, 2012), as well as the current pandemic of COVID-19 [1, 2]. COVID-19 is caused by a single-stranded, positive-sense RNA virus (SARS-CoV-2) that belongs to the *Coronaviridae* family [2]. The outbreak, first detected in Wuhan (China), was declared a pandemic by the World Health Organization (WHO) on 11 March 2020, as it rapidly spread globally [3]. As of 14 January 2021, over 108 million cases were reported worldwide, claiming over 2.3 million deaths [4]. There were 272,840 confirmed COVID-19 cases in Nepal and 2055 deaths [5].

Gastrointestinal (GI) symptoms are often common in patients infected with SARS-CoV-2, with one hospital in the US reporting 70% of GI patients testing positive for coronavirus [6–9]. Although the primary transmission source is through respiratory aerosol, studies confirmed fecal shedding [10, 11] and potential fecal-oral transmission of coronavirus [12, 13]. Since SARS-CoV-2 is shed through feces, it has also been detected in wastewater [14, 15]. Therefore, detecting the virus in sewage and wastewater can serve as an early detection method for identifying communities with circulating viruses in densely populated cities.

Environmental surveillance (ES) offers a complementary and more feasible approach to clinical disease surveillance. This approach yields population-level information and detects viral shedding by both symptomatic and asymptomatic patients [1, 16], thus providing a snapshot of the outbreak over an entire sewage catchment area and an early indication of clinical cases in the area [17]. Because not all symptomatic patients get tested due to reluctance or lack of access to tests, clinical samples may not be a sensitive indicator of cases in a given area [18, 19]. As experienced by many countries, clinical and community-based COVID-19 surveillance is expensive, technically challenging, and time-consuming, and therefore hard to implement in the manner needed to rapidly inform public health measures to contain the outbreak [10, 20, 21]. A longitudinal study conducted in Boston (USA) between March-April in 2020 showed a high correlation between environmental samples testing positive for the virus 4–10 days before symptoms presented in people in the sampled areas [22]. Additionally, wastewater sampling was found to be as sensitive as stool sampling for viral detection during vaccine-derived poliovirus outbreaks in Cuba [23]. As such, the versatility of environmental samples combined with the implementation of cheaper, user-friendly, and short turnaround sequencing tools such as MinION (Oxford Nanopore Technologies, UK) [23] may be invaluable tools used to rapidly identify viral strains and detect clusters of infection going forward.

Whole-genome sequencing (WGS) of viral pathogens is frequently used in disease surveillance and monitoring. The importance of genomic surveillance has been widely recognized as an important method of understanding viral evolution [24]. Portable and reliable sequencing tools such as MinION can play a crucial role in conducting genomic surveillance in a low-income country like Nepal, where there is limited access to high throughput DNA sequencing machines. The pocket-sized, field-deployable next-generation sequencer has enabled real-time outbreak surveillance of several recent outbreaks of Zika, Ebola, and Lassa viruses [1]. Obtaining whole or partial genome sequences of the virus during an outbreak is critical in understanding viral evolution and disease epidemiology and can help design accurate diagnostic tests for rapidly evolving RNA viruses such as SARS-CoV-2 [25]. RNA metabarcoding can also be utilized for the samples as it can overcome the limitations of conventional methods by targeting a variety of species and providing greater resolution of their taxonomies [26].

This study was conducted in selected areas of Kathmandu (Nepal) to evaluate the effectiveness of SARS-CoV-2 detection and characterization from environmental samples using

portable DNA sequencing technology (MinION). Studies in other countries have detected SARS-CoV-2 in environment samples collected from wastewater treatment plants (WWTP) [2, 16, 27–30]. However, Kathmandu has only one functional WWTP (out of five), and therefore, collecting all representative citywide sampling from WWTP is not possible. Since sewage lines were mapped in parts of the city for ongoing typhoid surveillance, the same sampling sites were used to conduct SARS-CoV-2 environmental surveillance. This cross-sectional study successfully detected, quantified, and characterized the SARS-CoV-2 virus from environment samples collected from the wastewater outlets and utility holes within catchment areas, providing vital information on the circulating strains of the virus in communities across Kathmandu.

## Materials and methods

### Study design and sewage sample collection

For initial feasibility and protocol validation, we carried out sewage sampling in three sites (TH-1, TH-2, and Te-3) in the Kathmandu district during June 2020 (Table 1). The catchment area of the three sites was chosen based on coverage and high population density.

To test our protocol after the validation and optimization in the preliminary sites, we selected 22 sewage catchment points from three wards (Ward 9 or Balkumari; Ward 11 or Sankhamul; and Ward 17 or Gwarko) of the Lalitpur Metropolitan City in the Lalitpur district. The district has a population of about 500,000 people [31]. In Balkumari, there were five sites; in Sankhamul, there were ten sites; and in Gwarko, there were seven sites (Table 2).

We adapted the field sampling strategy from our ongoing research on environmental surveillance of typhoid-causing bacterial pathogen *Salmonella typhi* from sewage. As part of the project, we conducted field surveys to comprehensively map sewage lines and the population size for each catchment area. We collected sewage samples from 22 selected catchment areas from July 26, 2020 to December 1, 2020 for the environmental surveillance of SARS-CoV-2 (Table 1).

Sewage samples were collected using an automated robotic pump (Biobot Analytics Inc., Cambridge, USA). For the environmental surveillance, grab (50 ml) samples were preferred based on outcome of optimization [32]. The probable reason for the consistent result in grab samples compared to the composite samples could be the dilution of viral particles by sewage discharge during non-viral peak hours [32]. The grab sewage samples were collected during the morning hour (7:00AM to 9:00AM) [33] based on the assumption that toilet activity and sewage discharge would be high during that time.

The samples were transported immediately to our Kathmandu-based laboratory using a cold chain (2–8°C) for processing. Each sample was pasteurized at 60°C for 90 minutes in a water bath [12] to inactivate any virus. Pasteurized samples were then subjected to differential centrifugation for virus segregation from sewage sludge by centrifuging at 3000 rpm for 30 minutes to pellet the bacterial cells and debris. The pellet was discarded, and the supernatant was precipitated for virus recovery using 15% PEG-6000 and 2% NaCl and gently shaken for 24 to 48 hours at 4°C [32]. The viral precipitates were then pelleted at 8000 rpm for 40 minutes,

**Table 1. Sample details of initial feasibility study.**

| S.N. | Sample ID | Ward | Tole | RT-PCR Rdrp | Metabarcoding |
|---|---|---|---|---|---|
| 1 | TH1 | | Sankhamul | Not detected | Detected |
| 2 | TH2 | | Thapathali | Not detected | Not detected |
| 3 | Te-3 | | Teku | Not detected | Detected |

**Table 2. Sample details of environmental surveillance in LMC.**

| S.N. | Sample ID | Ward | Tole | RT-PCR Rdrp | WGS_QC |
|---|---|---|---|---|---|
| 1 | BA-1 | 9 | Balkumari | Not detected | Fail |
| 2 | BA-2 | 9 | Balkumari | Not detected | Fail |
| 3 | BA-3 | 9 | Balkumari | Not detected | Fail |
| 4 | BA-4 | 9 | Balkumari | Not detected | Fail |
| 5 | BA-5 | 9 | Balkumari | Not detected | Fail |
| 6 | GW7 | 17 | Gwarko | Not detected | Fail |
| 7 | GW10 | 17 | Gwarko | Detected | Fail |
| 8 | GW11 | 17 | Gwarko | Detected | Pass |
| 9 | GW12 | 17 | Gwarko | Not detected | Pass |
| 10 | GW13 | 17 | Gwarko | Detected | Fail |
| 11 | GW14 | 17 | Gwarko | Detected | Pass |
| 12 | GW15 | 17 | Gwarko | Detected | Pass |
| 13 | CH4 | 11 | Sankhamul | Detected | Pass |
| 14 | KU1 | 11 | Sankhamul | Detected | Pass |
| 15 | SA1 | 11 | Sankhamul | Detected | Fail |
| 16 | SA2 | 11 | Sankhamul | Detected | Pass |
| 17 | SA3 | 11 | Sankhamul | Detected | Fail |
| 18 | SA4 | 11 | Sankhamul | Detected | Pass |
| 19 | SA5 | 11 | Sankhamul | Detected | Pass |
| 20 | SA6 | 11 | Sankhamul | Detected | Pass |
| 21 | SA7 | 11 | Sankhamul | Detected | Pass |
| 22 | SA9 | 11 | Sankhamul | Detected | Pass |

and the pellet was re-suspended in 400μl of SM buffer (Tris/NaCl/MgSO4) for further processing. The samples were handled in an enhanced Biosafety Level 2 laboratory with full personal protective equipment.

## SARS-CoV-2 detection using RT-PCR and estimation of prevalence rate

Initially, a diagnostic Real-Time polymerase chain reaction (PCR)-based assay (detecting envelope, nucleoprotein, and ORF1) was used to detect and quantify (sensitivity ~20 viral copies/ml) SARS-CoV-2.

RNA was extracted from the viral suspension (200μl) (abGenix viral DNA and RNA Extraction Kit, AIT Biotech, Singapore) in an automated nucleic acid extraction system (abGenix, AITBiotech, Singapore). We used Allplex™ RT-PCR SARS-CoV-2 Assay (Seegene Inc., Korea) to detect the presence of SARS-CoV-2 in sewage samples. The 5μl of extracted RNA was mixed with 10μl of a qPCR mix consisting of 2019-nCoV MOM (3μl), Real-time One-step Enzyme (1.2μl), 5X Real-time One-step Buffer (3μl) and Nuclease free water (2.8μl) with manufacturer recommended conditions- cDNA synthesis (50°C for 20 min); polymerase activation (95°C for 15 min); PCR (45 cycles of denaturation at 94°C for 15 s, and annealing/extension at 58°C for 30 s). The viral load data obtained was converted to RNA copies per liter of sewage first.

We estimated the Prevalence of SARS-CoV-2 within the catchment areas (community) using the mass balance Formula (1) on the total number of viral RNA copies in the wastewater each day as described by Ahemed et.al., 2020 [34].

$$Persons\ infected = ((sewage\ RNA) * (water\ per\ day)) / ((feces\ per\ day) * (viral\ shedding)) \quad (1)$$

We used *2019-nCoV_RdRp* (ORF1ab) Positive Control (cat. 10006897, IDT USA) obtained from the University of California-Davis with a known virus copy number to build a standard curve for determining the viral load of SARS-CoV-2. We then converted viral load data into RNA copies per liter of sewage. By utilizing this data, we estimated SARS-CoV-2 prevalence in sewage through a Monte Carlo approach executed in Oracle Crystal Ball (Release 11.1.2.4.600). The RNA concentration in feces excreted by an individual was estimation by using gamma distribution calculation (with parameters- shape = 10.78, scale = 0.46) [19]. A mean daily stool mass per person (75g with a standard deviation of 130.2) from low to middle-income countries was assumed as reported in Rose et al., 2015 [34, 35]. Using already available data for the population size of each catchment area, the prevalence rate per 1000 persons was calculated (minimum, mean, and maximum prevalence of SARS-CoV-2). By using the Heatmap plugin in QGIS v3.16 [36], we rendered the hotspots and clusters of infection prevalence across the sampling area in Gwarko and Sankhamul wards by determining a kernel density of each sampling point weighted from the mean estimated prevalence rate as input point vector layer. The larger number of clustered points results in a larger density of the identified hotspots.

## Detection and analysis of SARS-CoV-2 using metabarcoding in MinION

Only negative RT-PCR samples were screened with the modified nested PCR [37] protocol for corona viral family detection, including SARS-CoV-2, using primers sets (FWD1/RVS2 &FWD2/RVS1) and producing 328bp and 520bp PCR amplicons [38] (Fig 1). 8µl of RNA template was mixed with 1µl of random hexamers and 1 µl of dNTPs and incubated at 65˚C for 5 minutes. This mixture was combined with 10µl of synthesis mix consisting of 10X RT (2µl),

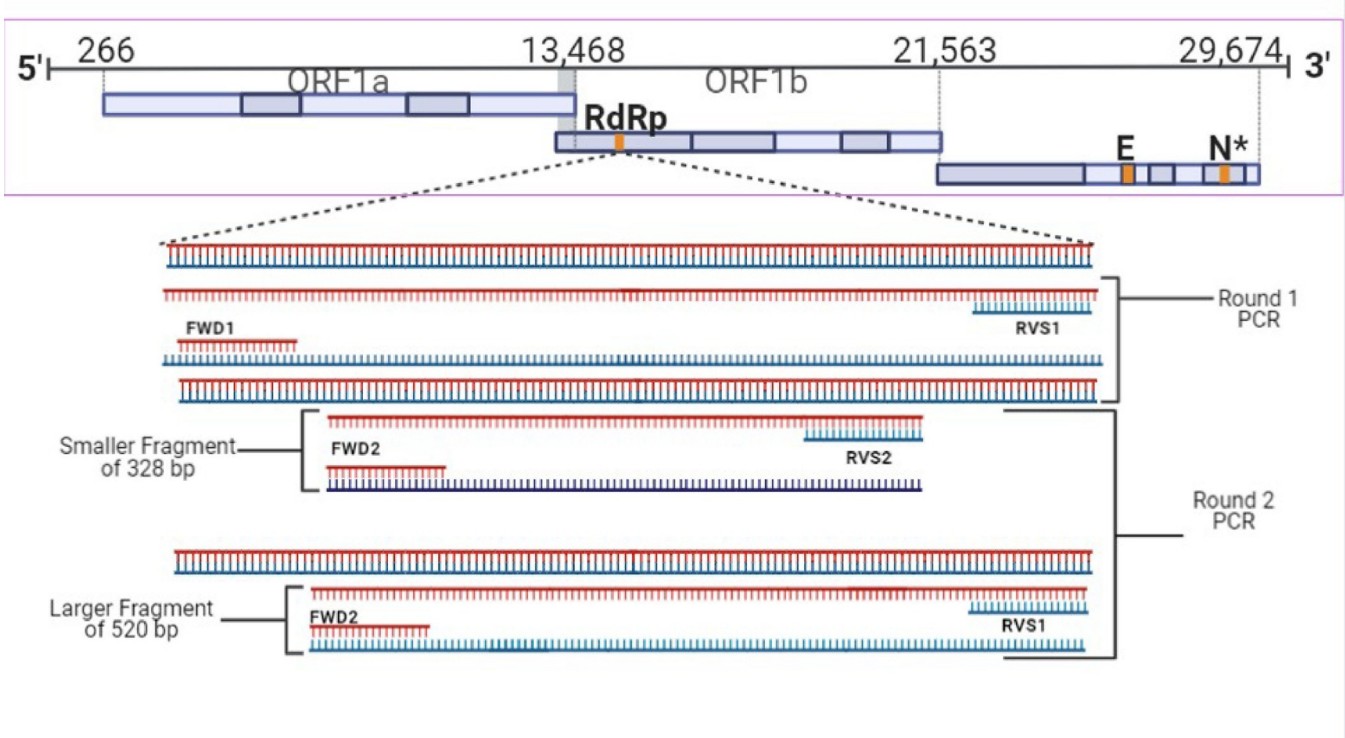

**Fig 1. Nested PCR to detect coronavirus using two sets of primers (FWD1/RVS2 & FWD2/RVS1).** Two PCR amplicons (328bp and 520bp) were produced, cleaned, and sequenced using Flongle MinION. The obtained sequences were BLAST analyzed in the NCBI database for viral taxonomic identification [3]. The images were created using the BioRender app.

25mM MgCl2 (4 μl), 0.1M DTT (2μl), RNase Out (1μl), and SSIII (1μl) (Superscript III (SSIII) cDNA synthesis kit, Invitrogen, USA). The final mixture was incubated at room temperature for 10 minutes, then incubated at 50˚C for 50 minutes, and then terminated at 85˚C for 5 minutes. The mixture was treated with 1μl of RNAse H and incubated at 37˚C for 20 minutes to remove any remnant RNA. The PCR amplicons were purified using a Montage Gel Purification kit (EMD Millipore Corp, USA) and sequenced using a portable next-generation sequencing device (Flongle MinION, Oxford Nanopore Technologies, UK). Nanoplot v1.30.1 [39] was used to check the quality of the raw nanopore fastq reads, and adapter sequences were trimmed using Porechop v0.2.4 [40]. Length and quality filtering were performed using Filtlong v0.2.0 [41], with minimum length and q-score thresholds of 300bp and seven. The cleaned sequence reads were then de-novo assembled using Canu v2.0 [42] to generate contigs with an overlap parameter threshold of 100bp. The scaffolds generated were subjected to the Basic Local Alignment Search Tool (BLAST) [41] for taxonomic identification against the locally downloaded GenBank database (Release 240, October 15 2020).

## Library preparation for whole-genome sequencing of SARS-CoV-2 from sewage

Tiled amplicon sequencing was used to amplify the whole genome of SARS-CoV-2 using ARTIC PCR protocol [43]. This protocol has been widely adopted to sequence the entire genome of SARS-CoV-2 from clinical samples. With 98 PCR primers, ARTIC PCR amplifies ~400bp amplicons spanning the SARS-CoV-2 genome with adjacent ~200bp overlaps. The sequencing library preparation is done by pooling these 98 primers in two pools and running two PCR reactions. We expected environmental samples to be highly degraded, and hence this protocol suits well for our study, as a smaller-sized amplicon increases the success of whole-genome amplification.

25μl reaction volume from each PCR pool was prepared containing 12.5μl Ampligold 360 2X MM, 3.6μl of primers, and 2μl template cDNA. The PCR conditions were: 95˚C for 30 secs followed by 25–45 cycles of 95˚C for 15 seconds and 60˚C for 5 mins. The ARTIC PCR amplicons were further cleaned (AMPure bead, Agencourt, Beckman Coulter, USA) and quantified (Qubit 3, Invitrogen Thermofisher Scientific, USA).

## Low-depth sequencing using Flongle in MinION

The PCR amplicons were pooled into a single tube at an equimolar concentration of 100fm. Native barcodes were assigned for each sample using Native barcoding expansion kit (EXP-NBD104) and ligation sequencing kit (SQK-LSK109) and run in Flongle Flowcell using MinION (Oxford Nanopore, UK). 14 samples were multiplexed in three FLO-FLG001 Flongle flow cells (R9.4.1) and sequenced on a MinION Mk1B device in three different sequencing runs. DNA library preparation had several quality control measures- ARTIC amplicons were quantified (after cleanup, barcode and adapter ligation) on agarose gel and a minimum of 200 fm DNA was used for library preparation, disqualifying all the low DNA samples. software (Oxford Nanopore, UK) was used to operate sequencing run on a Thinkpad P72 Mobile Workstation (Lenovo, USA). The raw data was generated and compiled, and real-time base calling was performed by the integrated production base caller Guppy v4.4.0 (Oxford Nanopore, UK) simultaneously during the sequencing run. Base calling was performed using high-accuracy mode (qscore threshold 7) within the operation software platform that generated fast5 and fastq reads. The RAMPART v1.2.0 [44] software package developed by the ARTIC network was used to monitor read mapping and genome coverage of SARSCoV2 in real-time.

### High-depth sequencing in Illumina MiSeq

ARTIC amplicons were cleaned using AMPure beads (Agencourt, Beckman Coulter, USA) and library were prepared using the Nextera XT library preparation and the Nextera indexing kits (Illumina, USA) [45]. The indexed library was then quantified with Qubit 3 (Thermoscietific, USA). The prepared library was sequenced on a Illumina Miseq using 300 cycle MiSeq Reagent Kit V2, and the data was de-multiplexed by MiSeq Reporter.

### Bioinformatics data analysis

The data was processed, and a consensus sequence was generated using a nextflow-based nfcore-viralrecon pipeline. The viralrecon bioinformatics analysis pipeline (https://github.com/nf-core/viralrecon) developed by a community-driven collaboration called nf-core was utilized to perform assembly and consensus generation of our collected SARS-CoV-2 samples. For the reads sequenced using the Illumina MiSeq, the pipeline performed adapter trimming using fastp version 0.23.2. The program, bowtie2 version 2.4.4, was then utilized to align and map the trimmed reads to a reference complete genome of SARS-CoV-2. The alignments were then sorted to remove low-quality reads using SAMtools version 1.14, and primer sequence removal from reads was performed using iVar version 1.3.1. Lastly, both BCFtools version 1.13 and Bedtools version 2.30 were used to call the consensus sequence for each sample. Similarly, the reads sequenced by the Oxford Nanopore Technologies (ONT) MinION sequencer of our samples were analyzed using the same analysis pipeline but using different programs. The pipeline performed read alignment and mapping to a reference SARS-CoV-2 genome using minimap2 version 2.24 and primer trimming and consensus building using Artic version 1.2.1 with the artic minion command, followed by the use of SAMtools version 1.14 to remove any unmapped reads.

### SARS-CoV-2 mutation and SNP analysis

The SARS-CoV-2 sequences obtained from the 12 wastewater samples were submitted for analysis in CoV-GLUE version 0.1.18 [46] to detect nucleotide mutations and coding region indels. This tool is based on a GLUE data-centric bioinformatics environment that analyzes nucleotide variations in user-submitted sequences of SARS-CoV-2, enabled by data from Epi-CoV of the GISAID database. The tool compares and analyses the sequences against the reference sequence of SARS-CoV-2 and generates lists of amino acid replacements and coding region indels. Geneious Prime 2020.2.3 and the bam-readcount tool v1.0.1 [47] were used to obtain the nucleotide frequency of the single nucleotide polymorphisms (SNPs) and deletions at the nucleotide positions reported in CoV-GLUE to validate the occurrence of the mutations or deletions and check for the possibility of multiple variants being found in a single sample. The variants across genomes of sequenced samples were visualized as bar plots generated using the R package ggplot2, with further annotation with Inkscape version 1.0.2–2.

## Results

### Detection of SARS-CoV2 in sewage and its prevalence estimation in the community

All the three preliminary sites had undetectable SARS-CoV-2. However, using metabarcoding approach, we detected SARS-CoV-2 and other coronaviruses, including Human coronavirus 229E, Human coronavirus NL63, duck-dominant coronavirus and rat coronavirus, in these four sites (S1 Table in S1 File).

Of 22 sewage samples collected during the environmental surveillance phase, we detected significant SARS-CoV-2 and determined viral load from 15 sites using the RT-qPCR test (S2 Table in S1 File). We did not detect the virus in all five sites of Ward 9/Balkumari and two sites of Ward 17/Gwarko. From the remaining 15 sites (5 in Gwarko and 10 in Sankhamul), we calculated an average of 40,613.83 per ml (range: 295 to 242,000) viral load. Among five sites in Gwarko, we calculated an average 72,712.5 per ml viral load. Similarly, among ten sites in Sankhamul, we calculated an average 24,564.5 per ml viral load (S2 Table in S1 File).

In case of the estimated mean SARS-CoV2 infection rate through sewage viral load, the catchment sites of Sankhamul with about 4,632 residents had an estimated prevalence of 3.5 (range: 0.19–30.96) per 1000 persons, while the catchment sites of Gwarko with 13,399 persons had estimated prevalence of 0.6 (SD: 0.04–5.09) per 1000 persons (S2 Table in S1 File and Figs 2 and 3). A site in Sankhamul, SA2, had the highest estimated prevalence of 11.36 (range: 062–100.49) per 1000 persons, while a site, GW15, in Gwarko had the lowest estimated prevalence of 0.11 (range: 0.01 to 0.97) per 1000 persons.

## Genome recovery of SARS-CoV-2 from sewage

14 samples passed initial QC in library preparation steps, of which only 12 yielded enough quantifiable PCR amplicons after ARTIC PCR. These amplicons were further processed for DNA sequencing on two platforms- Oxford Nanopore MinION and Illumina MiSeq. An average depth of 16.9x and breadth from 34.7% to 86.2% genome coverage were obtained in 12 samples using Flongle in MinION. Similarly, through Illumina MiSeq, we obtained the consensus genomes at an average depth of 240x and breadth from 29% to 99% genome coverage. A pairwise comparison of the consensus genome sequences generated from two sequencing platforms showed they were highly identical (99.96% to 100% identity) (S8 Table in S1 File).

All of the consensus sequences obtained from MinION were compared against the reference sequence (accession no. MN908947.3) using the CovGlue web application. **B.1** (16.7%), **B.1.36** (33.3%) and **B.1.1** (50.0%) were identified as circulating lineages of SARS-CoV-2 in the environmental (sewage) samples (Table 3).

## Mutation analysis and SNP frequency detected in SARS-CoV-2 genomes

We detected a total of 47 unique nucleotide mutations across 12 sequenced samples, out of which more than half were non-synonymous (n = 23) and deletions (n = 10), and 14 were silent mutations using Nanopore MinION (S3 Table in S1 File). The spike D614G was the most frequent mutation observed in all samples, while we observed other missense mutations such as S477N, A570S, and K1073N in the spike gene of one sample from Sankhamul. A frameshift deletion in the spike gene of a sample, SA6, from Sankhamul was also detected. Another gene, ORF3a, adjacent to the spike also had major missense and deletion mutations altering the protein sequence or structure across ten samples. Other frequent missense mutations detected were V381A in NSP2, M86I in NSP6, P314L in NSP12b, Q57H, and V77I in ORF3a genes. Overall, we observed an average of 7.8 mutations per sample with a maximum of 14 and a minimum of 3 mutations in a sample.

Using Illumina MiSeq, we detected 211 unique nucleotide mutations across ten sequenced samples. Almost 60% of the mutations were non-synonymous (n = 126), while we detected two nonsense and 13 deletion mutations. Out of the total mutations, 70 (33%) were synonymous and did not alter amino acids in a protein sequence. We detected spike D614G in seven samples, while three samples did not have any reads in the corresponding region of the genome. We also observed other missense mutations (S477N, A570S, and K1073N) in the spike gene from MinION. In addition to the mutations observed in the ORF3a gene from

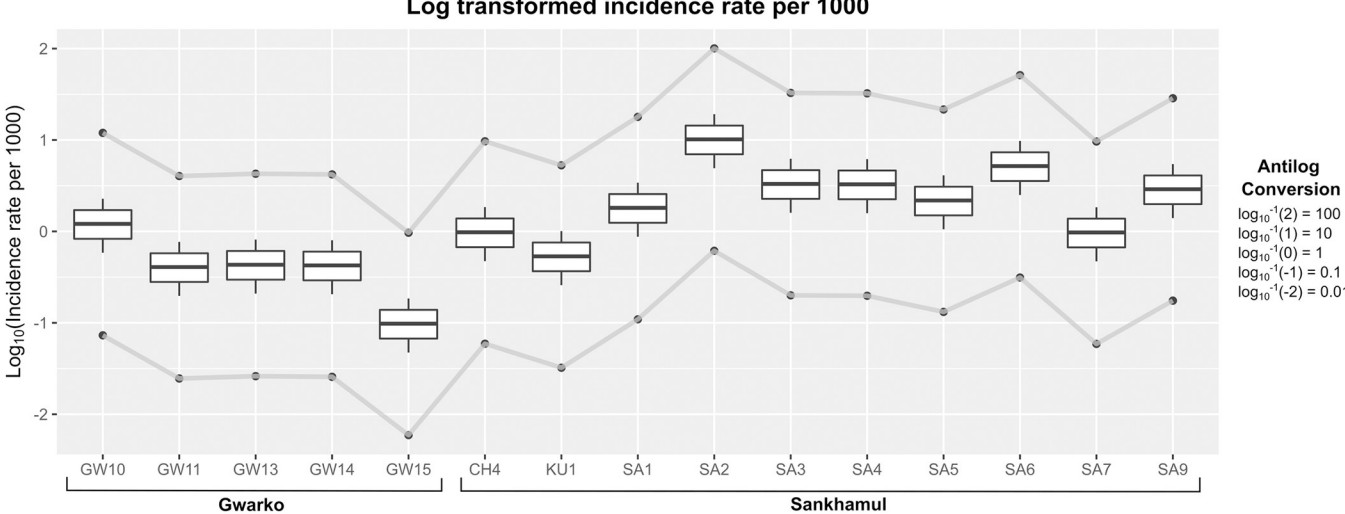

**Fig 2. Mean prevalence (in log scale) of Covid-19 per 1000 persons in various catchment areas of Sankhamul and Gwarko.** It also highlights the maximum and minimum prevalence (line graph) in the catchment areas. Conversion of the log values as per the prevalences of Covid-19 per 1000 persons is shown in the legend.

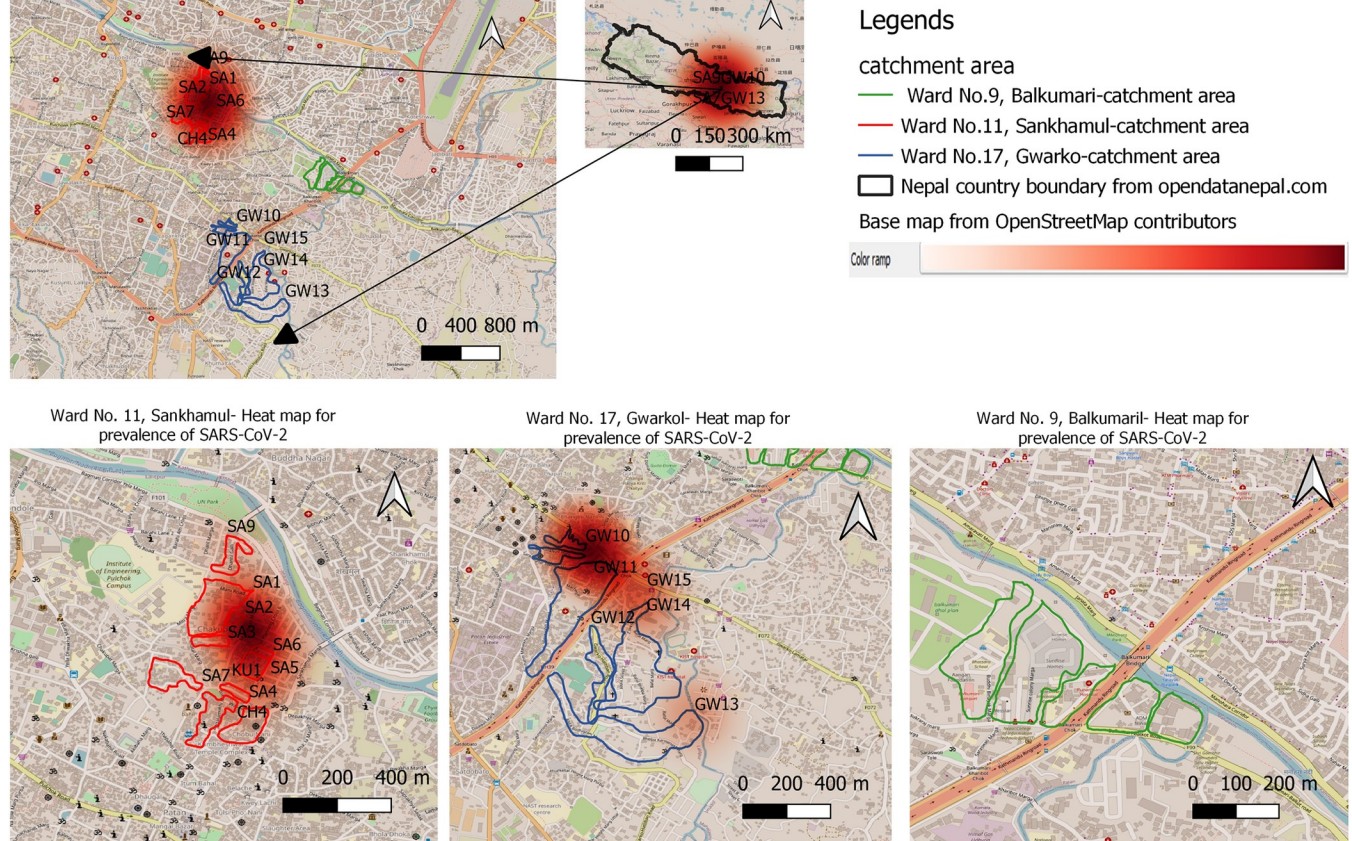

**Fig 3. SARS-CoV-2 infection heat-map of three sampled wards (11, 9, and 17) of the Lalitpur Metropolitan City (Kathmandu valley) with predicted SARS-CoV-2 prevalence per 1000 people in sewage samples.** The map was created using QGIS v3.16 s [36]]with a base map from the *OpenStreetMap* contributors. The QGIS heat map plugin assigned the gradient with a max score of one and a minimum of 0.

**Table 3. SARS-CoV-2 Pangolin lineages of consensus sequence generated from the environmental samples from CovGlue submission.**

| S.N | SAMPLING SITE | PANGOLIN LINEAGE | S.N | SAMPLING SITE | PANGOLIN LINEAGE |
|---|---|---|---|---|---|
| 1 | SA2 | B.1.36 | 9 | GW11 | B.1.1 |
| 2 | SA7 | B.1.1 | 10 | GW12 | B.1.36 |
| 3 | CH4 | B.1.36 | 11 | GW14 | B.1.1 |
| 4 | KU1 | B.1 | 12 | GW15 | B.1 |
| 5 | SA6 | B.1.36 | 13 | GW13 | NA |
| 6 | SA5 | B.1.1 | 14 | SA3 | NA |
| 7 | SA4 | B.1.1 | 15 | GW10 | NA |
| 8 | SA9 | B.1.1 | 16 | SA1 | NA |

MinION, we detected additional mutations in V55G, T223I, and P267L that would alter its protein sequence and structure. Most of the mutations observed from the MinION were also observed using MiSeq, albeit in much higher depth and coverage (S7 Table in S1 File).

An average of 142,692 reads per sample were produced using MiSeq compared to 1806 in MinION. Sequence quality of high depth Illumina Miseq is greater and extensive than that of MinION. However, sequencing from both platforms yielded the detection of mutations of interest, including D614G, P314L, and S477N.

The SNP frequency analysis (from both platforms) showed varying nucleotide frequencies on the SNP positions highlighted by the CoVGlue. For example, the frequency analysis of the mutation V381A, on the NSP1 gene of the MinION sequenced SA2 sample showed 42.60% cytosine (C) and a 50.80% thymine (T)- indicating almost a 50–50% chance of amino acid changing nucleotide mutation. Other mutations were observed to show similar nucleotide frequency patterns (S1a, S1b Fig and S3 and S9 Tables in S1 File). With different variant having main characterizing mutations that defines the variant (mostly focused on S-gene- S1 Fig in S1 File) different variant could be classified. A wide distribution of SNPs with varying abundance were observed in the **vcf** file generated by the ARTIC pipeline. The SNP-based analysis, where important mutations of concern (such as mutations in the S gene) provided genetic/variant diversity status of the virus (S1 Fig in S1 File).

## Discussion

As demonstrated by its successful application in the global efforts to eradicate poliovirus [24], ES can be an important supplement to clinical surveillance. In particular, ES can contribute to several critical areas, such as characterizing regional diversity in areas without clinical surveillance, mapping geographic micro-diversity to inform risk-based interventions, early detection of outbreaks before clinical presentation, and advancing real-time assessment of interventions to inform the need for additional public health measures [24]. ES can be inexpensive and fast [48] and like clinical surveillance, it can offer multiple sources for samples collection apart from sewage, like river system, [49], air [50], or even screening for evidences of other indicator pathogens [48]. During an outbreak of typhoid fever in Nepal, water isolates had over 96% analytical similarity to blood culture results [51].

We found only one additional study conducted in Nepal pertaining to detection of SARS-CoV-2 via environmental sampling. [52]. The study reported presence of the coronavirus from multiple sources of environmental sampling: wastewater treatment plants, hospital wastewater, river water, and sewage. Although, only two sewage samples were analyzed in the study, SARS-CoV-2 was detected in all four qPCR assays that were tested confirming viability of

environmental (sewage) sampling. Only hospital wastewater had similar success rate and other sources of the samples did not.

Rapid detection of SARS-CoV-2 in environmental samples (such as sewage) and viral strain characterization using portable genomic tools provide important information on disease distribution at a community level [22, 24]. Such data can help identify transmission hotspots in urban and rural communities and provide information on circulating strains of viruses, as evident in S1 Table in S1 File. Similarities in estimated prevalence per 1000 capita obtained from our sampling in Gwarko and reported average within the same period highlight an accurate depiction of infection rate within the community. Alternatively, a higher mean prevalence rate in Shankhamul sites compared to the reported average within the same period can allude to the identification of an infection hotspot in that region. It must be noted that in our study, SARS-CoV-2 was detected from grab samples only. The other study conducted in Nepal suggested using 24-hour composite samples to understand variability of the viral RNA concentration [52] but our efforts to do the same yielded in highly diluted samples with very negligent amount of viral RNA present in those samples.

In this study, we implemented and optimized a system of detecting the virus from sewage samples and, based on viral load, predicted the prevalence rate using Monte Carlo simulation from which a heat map of SARS-CoV-2 in the selected areas of the Kathmandu valley was constructed. This was a novel approach to estimate prevalence of cases in a possible hotspot scenario using only real-time PCR data (including viral load calculations). In resource-limited, low-income countries like Nepal, access to tools and technologies required to perform genomic-based detection and surveillance can be limited. This feasibility study has demonstrated that ES based on sewage samples using highly sensitive nested PCR combined with portable next-generation sequencing technology can provide adequate sensitivity, specificity, and sequencing depth to detect, quantify, and characterize SARS-CoV-2 in the community.

Whole-genome sequencing of the SARS-CoV-2 was obtained using ARTIC primers in MinION (Oxford Nanopore Technologies, UK). By using a meta-barcoding approach on RT-qPCR negative samples along with SARS-CoV-2 [26], we were able to detect other coronaviruses, such as human coronavirus 229E, a known cause of the common cold with global distribution [24, 53]. The presence of duck-dominant coronavirus and rodent coronavirus in sewage suggested an abundant mixture of various coronaviruses originating from other species. Detection of various coronaviruses in sewage samples using this approach demonstrated the potential of using this technique to detect and characterize new and emerging coronaviruses among species in close proximity, especially given the propensity for coronaviruses to recombine [54].

Out of 22 sampled sites, 16 had SARS-CoV-2 present. The heat-map showed potential hotspots for SARS-CoV-2 prevalence. Analyzing enough sampling points can provide a good picture of circulating viruses and predict hotspots in a community with greater resolution and accuracy. Further studies can be designed to look into prevalence estimation in communities and correlate that with available clinical data.

SARS-CoV-2 extracted from wastewater samples was successfully sequenced with just enough depth and coverage with the Oxford Nanopore Technologies (ONT) MinION long-read sequencer. Our results support the use of the MinION device as a cost-effective, rapid, and accurate sequencing tool to provide sequencing opportunities to personnel and researchers in remote areas or institutions that may not have access to the funds to purchase short-read NGS platforms (e.g., Illumina, Ion Torrent), with the low-priced MinION device able to sequence the SARS-CoV-2 genome with accuracy (S8 Table in S1 File), albeit with lower depth compared to the sequences generated by the Illumina MiSeq device (S7 Table in S1 File). Therefore, we suggest using the MinION device as an effective preliminary sequencing tool to

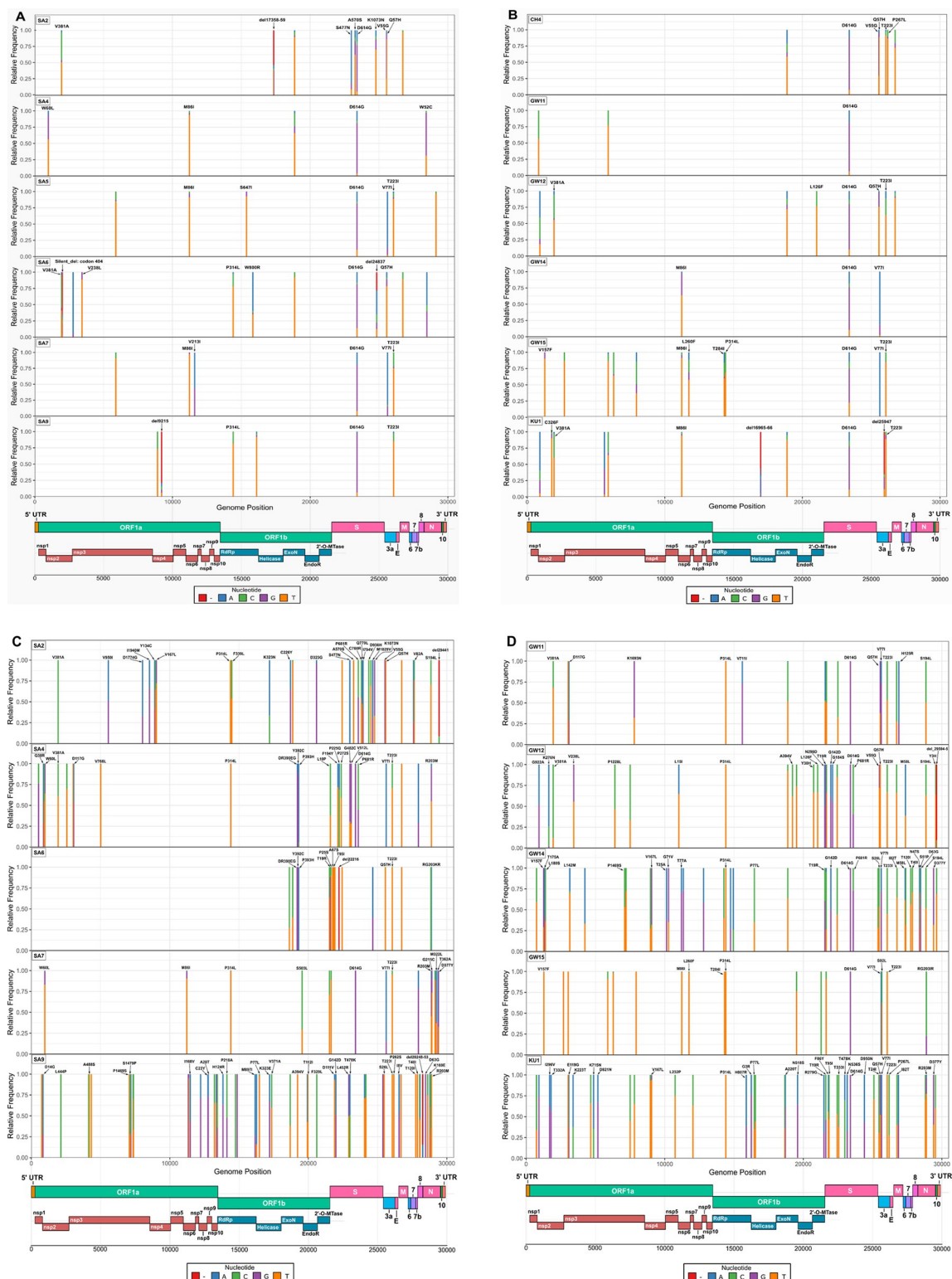

**Fig 4. Key Distribution of mutations, both synonymous and nonsynonymous, including nucleotide deletions across the full-length genomes of SARS-CoV-2 from wastewater samples collected in this study.** Nucleotide frequencies at each nucleotide position of the mutations are indicated with different colors (shown in key) to show nucleotide variation in the reads generated during sequencing, obtained using Geneious Prime 2020.2.3. (A, B) indicates nucleotide frequency of mutations, generated during Nanopore MinION sequencing of the 12 wastewater samples collected in the study. (C, D) indicates nucleotide frequency of mutations, generated

during Illumina MiSeq sequencing of 10 of the wastewater samples. The SARS-CoV-2 genome organization is schematically represented on the bottom of the plots. Amino acid mutations, SNPs and deletions have been annotated on the plot according to the presence of significant nucleotide frequency changes on each position, with the exclusion of labels for synonymous nucleotide mutations.

obtain accurate, low-depth sequences quickly in pandemic condition, followed by the use of short-read sequencers such as the Illumina MiSeq to produce sequences with high depth as a form of verification, as adapted by our study (Fig 4).

The obtained consensus genome sequences of SARS-CoV-2 from the sewage samples identified **B.1**, **B.1.36,** and **B1.1** as the most probable major circulating lineages (Table 3, S2 Table in S1 File). Whole-genome data of SARS-CoV-2 from 14 clinical samples were sequenced at our facility for the Nepal Health Research Council (Government of Nepal) inferred from the GISAID database. Three samples originate from the Kathmandu valley, and identified lineages include **B.1.36** and **B.1.130**. The detected SARS-CoV-2 lineages (**B.1, B.1.1, B.1.36**) and key mutations such as **D614G** in the clinical samples were also circulating in the sewage samples. Some samples with low genomic coverage failed to assign lineage beyond B.1.

Novel mutations with possible clinical significance were also detected- such as frameshift and silent deletion mutations in the **S**, **NSP13**, **ORF3a,** and **NSP2** genes. These novel mutations during data analysis were yet to be reported in the global database. Additionally, a determination regarding their functional characterization is needed. For example, a frameshift deletion in the S gene could have important clinical and epidemiological significance. The D614G (in the S gene) was spontaneously detected in almost every sample, a defining mutation for the G clade [55]. This particular mutation was associated with increased infectivity of SARS-CoV-2. Another detected mutation, S477N (in the S gene), was reported to increase the binding affinity of the virus to the host's angiotensin-converting enzyme 2 (ACE-2) receptor giving higher transmission capability [56, 57]. Similarly, novel missense mutations detected in **NSP12**, **NSP2,** and **N** genes from sites in the same ward indicated probable ongoing viral evolution in the community. This finding also highlighted the need for genomic surveillance to track viral evolution and emerging variants. Though the environmental samples might have more than one variant circulating, which could be detected as SNP in genomic analysis. SNP analysis revealed possibility of assessing circulating major/minor variant diversity on environmental samples based on key mutations. (S3 Table in S1 File).

The detection and assessment of circulating viral variants (SARS-CoV-2) using genomic surveillance can help evaluate the effectiveness of vaccination (immunization) efforts- something high-income countries like the United Kingdom has realized from the early onset of COVID19, and hence was actively contributing information to the global genomic database for SARS-CoV-2 [58]. In contrast, low-income countries like Nepal struggled to understand COVID-19 prevalence and transmission due to a lack of genomic laboratory capabilities [59]. Genomic surveillance using environmental samples as presented in this study could be a possible solution for resource-strapped countries to estimate SARS-CoV-2 presence at the community level and determine the circulating strains of the virus that could be used to devise the most appropriate interventions. Clinical surveillance coupled with environmental surveillance could be the most effective way to prevent and manage outbreaks.

## Supporting information

**S1 File. All supplementary figures and tables.**
(DOCX)

## Acknowledgments

We would like to thank the Mayor's Office of the Lalitpur Metropolitan City for providing the permit and encouraging us to conduct this research. We would like to thank Professor Eric Alm and Dr. Noriko Endo of BIOBOT (USA) for providing composite sampling robots. Finally, we would like to thank everyone at Intrepid Nepal Pvt Ltd, Center for Molecular Dynamics Nepal, and BIOVAC Nepal for their tireless work even during the nationally enforced lockdown in Nepal.

## Author Contributions

**Conceptualization:** Rajindra Napit, Dibesh B. Karmacharya.

**Data curation:** Rajindra Napit, Prajwol Manandhar, Ashok Chaudhary, Bishwo Shrestha, Ajit Poudel, Roji Raut, Saman Pradhan, Samita Raut, Rajesh M. Rajbhandari.

**Formal analysis:** Rajindra Napit, Prajwol Manandhar, Rajesh M. Rajbhandari.

**Funding acquisition:** Rajesh M. Rajbhandari, Dibesh B. Karmacharya.

**Investigation:** Rajindra Napit, Ashok Chaudhary, Bishwo Shrestha, Roji Raut, Saman Pradhan, Samita Raut, Anupama Gurung.

**Methodology:** Rajindra Napit, Prajwol Manandhar, Ashok Chaudhary, Roji Raut, Saman Pradhan, Samita Raut, Anupama Gurung.

**Project administration:** Rajindra Napit, Ashok Chaudhary, Bishwo Shrestha, Rajesh M. Rajbhandari.

**Resources:** Rajesh M. Rajbhandari, Christine K. Johnson, Jonna K. Mazet, Dibesh B. Karmacharya.

**Software:** Rajindra Napit, Prajwol Manandhar, Pragun G. Rajbhandari, Rajesh M. Rajbhandari.

**Supervision:** Sameer M. Dixit, Jonna K. Mazet, Dibesh B. Karmacharya.

**Validation:** Rajindra Napit, Prajwol Manandhar, Ajit Poudel, Pragun G. Rajbhandari, Rajesh M. Rajbhandari, Dibesh B. Karmacharya.

**Visualization:** Rajindra Napit, Prajwol Manandhar, Pragun G. Rajbhandari.

**Writing – original draft:** Rajindra Napit, Prajwol Manandhar, Ajit Poudel, Roji Raut, Pragun G. Rajbhandari, Dibesh B. Karmacharya.

**Writing – review & editing:** Rajindra Napit, Prajwol Manandhar, Ajit Poudel, Rajesh M. Rajbhandari, Jessica S. Schwind, Christine K. Johnson, Jonna K. Mazet, Dibesh B. Karmacharya.

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
