## [Decision Letter · Decision Letter 0]

26 Oct 2022

PONE-D-22-13706Rapid genomic surveillance of SARS-CoV-2 in a dense urban community using environmental (sewage) samplesPLOS ONE

Dear Dr. Karmacharya,

Thank you for submitting your manuscript to PLOS ONE. After careful consideration, we feel that it has merit but does not fully meet PLOS ONE’s publication criteria as it currently stands. Therefore, we invite you to submit a revised version of the manuscript that addresses the points raised during the review process.

We look forward to receiving your revised manuscript.

Kind regards,

Nadim Sharif, M.Sc.

Academic Editor

PLOS ONE

Journal Requirements:

“We would like to thank the Mayor’s Office of the Lalitpur Metropolitan City for providing the permit and encouraging us to conduct this research. We would like to thank Professor Eric Alm and Dr. Noriko Endo of BIOBOT (USA) for providing composite sampling robots. We would also like to express our gratitude to the One Health Institute of the University of California-Davis and the USAID-funded PREDICT project for providing us with laboratory resources. Whole-genome sequencing was done at the Intrepid Nepal Genomic Center. Some of our work was partially funded by the Australian Development Agency and PSI grant (the Netherlands). We would like to thank all these agencies for their support. And finally, we would like to thank everyone at Intrepid Nepal Pvt Ltd, Center for Molecular Dynamics Nepal, and BIOVAC Nepal for their tireless work even during the nationally enforced lockdown in Nepal.”

“University of California-Davis and the USAID-funded PREDICT project for providing us with laboratory resources. Whole-genome sequencing was done at the Intrepid Nepal Genomic Center. Some of our work was partially funded by the Australian Development Agency and PSI grant (the Netherlands)”

4. We note that Figures 1 & 5 in your submission contain [map/satellite] images which may be copyrighted. All PLOS content is published under the Creative Commons Attribution License (CC BY 4.0), which means that the manuscript, images, and Supporting Information files will be freely available online, and any third party is permitted to access, download, copy, distribute, and use these materials in any way, even commercially, with proper attribution. For these reasons, we cannot publish previously copyrighted maps or satellite images created using proprietary data, such as Google software (Google Maps, Street View, and Earth). For more information, see our copyright guidelines: http://journals.plos.org/plosone/s/licenses-and-copyright.

 a. You may seek permission from the original copyright holder of Figures 1 & 5 to publish the content specifically under the CC BY 4.0 license. 

Additional Editor Comments (if provided):

1. Remove figures 1-3 from result section to method section.

2. Revise the supplementary document and represent the mutation in more acceptable way example, T358C

3. Follow the author guidelines of PLOS ONE to format the article. Make sure the references are used in the required styles.

Reviewers' comments:

Reviewer's Responses to Questions

**Comments to the Author**

1. Is the manuscript technically sound, and do the data support the conclusions?

Reviewer #1: Partly

Reviewer #2: Partly

2. Has the statistical analysis been performed appropriately and rigorously? 

Reviewer #1: Yes

Reviewer #2: N/A

3. Have the authors made all data underlying the findings in their manuscript fully available?

Reviewer #1: Yes

Reviewer #2: Yes

4. Is the manuscript presented in an intelligible fashion and written in standard English?

Reviewer #1: Yes

Reviewer #2: Yes

5. Review Comments to the Author

Reviewer #1: In the present manuscript PONE-D-22-13706 "Rapid genomic surveillance of SARS-CoV-2 in a dense urban community using environmental (sewage) samples", the authors have studied/presented the genomic surveillance of SARS-CoV-2 in wastewater samples Kathmandu, Nepal, by two different procedures, MiSeq (Illumina) and MinION (ONT). This is a well-written manuscript, the scope of which is really important in the current pandemic, and more important in countries like Nepal, as stated by the authors.

As I can see, 4 reviewers have also evaluated the manuscript, and this is a revised or resubmitted manuscript. Although, I have not seen the initial submission, the authors have tried to address the comments/critishisms of the reviewers to some extent.

Major comments

1. The results of the study are not well presented, not really validated and moreover not novel for the readers.

2. In this regard, the fig. 1-3 does not include any results but describing the sample sites/plan and the basic analytical procedures.

3. Fig. 6, including the mutations/SNPs detected is poor, given the fact that WGS was performed.

4. The authors must present the %percentage of the mutations/SNPs detected and moreover to translate them to specific SARS-CoV-2 variants.

5. Several discrepancies between MiSeq and MinION are presented. This is very important, and the authors should discuss it in detail.

6. The authors should describe in detail the quality control of the NGS libraries and the final sequencing output.

7. The quantification of viral load in WW samples and the "persons infected" formula are not clearly presented and described.

8. The authors should pre

9. And most importantly, I cannot draw any specific and major conclusions from the results and discussion presented.

Reviewer #2: Napit et al. studied the genomic surveillance of SARS-CoV-2 in sewage from Kathmandu, Nepal. The authors aimed to establish a genomic-based environmental surveillance system by monitoring SARS-CoV-2 in sewage samples via PCR and WGS approaches. Although the experiments in the manuscript are well-organized and the manuscript itself is also written in a good way, this reviewer can see right away the shortages of this study. The following are listed some major issues for the authors' consideration.

1) One of the main issues is the data quality of the sequencing. I looked into some of the raw reads the authors deposited on GitHub. Since those data are directly related to the conclusions, it will not stand if such bad quality data was used. The author should consider toning down some of the conclusions from Fig. 6.

2) Considering the environmental variants which might cause damage to lots of biological samples, the mutations the authors found has simply after the virus particles expose to sewage. It is essential to show some correlation to some clinical results to see if the mutations were presented in society.

3) This reviewer really appreciates this kind of excellent effort in studying SARS-CoV-2 in developing countries like Nepal. The authors can discuss the significance of this study extensively since the novelty of this study is not apparent, especially in the methodology and results. One possible direction is to mention that sewage might become one of the possible infection routes for SARS-CoV-2, and we should be cautious since some mutations were found in environmental samples, such as sewage, in this study.

6. PLOS authors have the option to publish the peer review history of their article (what does this mean?). If published, this will include your full peer review and any attached files.

Reviewer #1: No

Reviewer #2: No

---

## [Author Response · Author response to Decision Letter 0]

1 Dec 2022

Reviewer 1 

In the present manuscript PONE-D-22-13706 "Rapid genomic surveillance of SARS-CoV-2 in a dense urban community using environmental (sewage) samples", the authors have studied/presented the genomic surveillance of SARS-CoV-2 in wastewater samples Kathmandu, Nepal, by two different procedures, MiSeq (Illumina) and MinION (ONT). This is a well-written manuscript, the scope of which is really important in the current pandemic, and more important in countries like Nepal, as stated by the authors.

 SNo Comment Response

 1 The results of the study are not well presented, not really validated and moreover not novel for the readers.

 The result section has been updated with more elaborate figure from supplementary data (figure 6) and table no. 3 has been added to make it more effective.

 2 In this regard, the fig. 1-3 does not include any results but describing the sample sites/plan and the basic analytical procedures. These figures are intended to elaborate the procedures/methodology and not the results- they are in methodology section.

 3 Fig. 6, including the mutations/SNPs detected is poor, given the fact that WGS was performed.

 The figure has been removed with the supplementary figure S1 which is more elaborate and depicts all the SNP with its relative abundance in y-axis while SNP position in x-axis.

 

 4 The authors must present the % percentage of the mutations/SNPs detected and moreover to translate them to specific SARS-CoV-2 variants. The mutation % has been added in supplementary table S3 and S9, Major circulating variant has been added in line 266-269 and table 3. The result is more specifically highlighted in the result section as well (line 301-304, 359-364 and 296-300). At the time of this study, variant of concerns was not classified and tools like Frejya that uses similar SNPs frequency-based algorithm and database containing all variant of concerns can actually predict circulation variant fractions (https://github.com/andersen-lab/Freyja) was not available.

 5 Several discrepancies between MiSeq and MinION are presented. This is very important, and the authors should discuss it in detail. The discrepancies has been highlighted both in result section line 292-293 Supplementary table S7.

 6 The authors should describe in detail the quality control of the NGS libraries and the final sequencing output The quality control has been added in line 190-193 and 202-206.

 7 The quantification of viral load in WW samples and the "persons infected" formula are not clearly presented and described. The formula was taken from the previous similar research paper; all the procedure was already described in those literature so was not mentioned earlier. The explanation has been added in line 135-137.

 

 8 And most importantly, I cannot draw any specific and major conclusions from the results and discussion presented.

 The discussion has been updated to add some clarity, we intended to demonstrate and highlight importance of genomic surveillance for disease like SARS-CoV-2. However, it is not feasible always for country like Nepal where fundings are not readily available, in such condition using low-cost sequencing to assess the situation and then if something important comes up then go for Miseq or clinical genomic surveillance of the community would be better option. Described in line 354-370, 380-392.

 

Reviewer 2 Napit et al. studied the genomic surveillance of SARS-CoV-2 in sewage from Kathmandu, Nepal. The authors aimed to establish a genomic-based environmental surveillance system by monitoring SARS-CoV-2 in sewage samples via PCR and WGS approaches. Although the experiments in the manuscript are well-organized and the manuscript itself is also written in a good way, this reviewer can see right away the shortages of this study. The following are listed some major issues for the authors' consideration.

 SNo Comment Response

 1 One of the main issues is the data quality of the sequencing. I looked into some of the raw reads the authors deposited on GitHub. Since those data are directly related to the conclusions, it will not stand if such bad quality data was used. The author should consider toning down some of the conclusions from Fig. 6. All the data was not of bad quality, few had low sequencing coverage, which is common in environmental samples as they do not have high viral load to that of clinical samples that has been acknowledged with all the table on coverage and depths in supplementary table and also in the manuscripts. The figure 6 has also been replaced with the Supplementary figure S1. Those low coverage samples had failed to distinguish lineage to high resolution (table no 3 has been added) for clarity. A line has been added in discussion highlighting the same (line 370-371)

 2 Considering the environmental variants which might cause damage to lots of biological samples, the mutations the authors found has simply after the virus particles expose to sewage. It is essential to show some correlation to some clinical results to see if the mutations were presented in society.

 The comparison with clinical samples were in discussion section but result was updated to contain variant calling data from consensus sequences in result section and table no. 3. 

 

 3 This reviewer really appreciates this kind of excellent effort in studying SARS-CoV-2 in developing countries like Nepal. The authors can discuss the significance of this study extensively since the novelty of this study is not apparent, especially in the methodology and results. One possible direction is to mention that sewage might become one of the possible infection routes for SARS-CoV-2, and we should be cautious since some mutations were found in environmental samples, such as sewage, in this study.

 The discussion (line 385-390 and 392-395) has been updated to highlight the effort needed to track pandemic in country like Nepal cheaply and quickly in time.

---

## [Editor Report · Decision Letter 1]

4 Dec 2022

PONE-D-22-13706R1Rapid genomic surveillance of SARS-CoV-2 in a dense urban community using environmental (sewage) samplesPLOS ONE

Dear Dr. Karmacharya,

Thank you for submitting your manuscript to PLOS ONE. After careful consideration, we feel that it has merit but does not fully meet PLOS ONE’s publication criteria as it currently stands. Therefore, we invite you to submit a revised version of the manuscript that addresses the points raised during the review process.

We look forward to receiving your revised manuscript.

Kind regards,

Nadim Sharif, M.Sc.

Academic Editor

PLOS ONE

Journal Requirements:

Additional Editor Comments:

Figure 1, 2 and 5 are not necessary for the main documents. Actually these figures reduced the quality of the manuscript.

For the map figures, the authors should use high resolution and detailed maps of the study regions. It seems the maps are copied from google map or similar application. It is totally unacceptable. Use your originally created figures to use in the manuscript. Figure 2 is not necessary in this manuscript. Remove figure 2 and improve figures 1 and 5.

Specify the reference genome sequence accession number against which you aligned your sequences and compared form mutational analysis.

The discussion section requires major improvement. Compare your findings with similar studies in your country and other countries and describe the similarities and differences with appropriate reasoning.

---

## [Author Response · Author response to Decision Letter 1]

9 Dec 2022

Reviewer 1 

In the present manuscript PONE-D-22-13706 "Rapid genomic surveillance of SARS-CoV-2 in a dense urban community using environmental (sewage) samples", the authors have studied/presented the genomic surveillance of SARS-CoV-2 in wastewater samples Kathmandu, Nepal, by two different procedures, MiSeq (Illumina) and MinION (ONT). This is a well-written manuscript, the scope of which is really important in the current pandemic, and more important in countries like Nepal, as stated by the authors.

 SNo Comment Response

 1 The results of the study are not well presented, not really validated and moreover not novel for the readers.

 The result section has been updated with more elaborate figure from supplementary data (figure 6- now figure 4) and table no. 3 has been added to make it more effective.

 2 In this regard, the fig. 1-3 does not include any results but describing the sample sites/plan and the basic analytical procedures. These figures are intended to elaborate the procedures/methodology and not the results- they are in methodology section. Still figure 1 and 2 has been removed.

 3 Fig. 6, including the mutations/SNPs detected is poor, given the fact that WGS was performed.

 The figure has been removed with the supplementary figure S1 which is more elaborate and depicts all the SNP with its relative abundance in y-axis while SNP position in x-axis.

 

 4 The authors must present the % percentage of the mutations/SNPs detected and moreover to translate them to specific SARS-CoV-2 variants. The mutation % has been added in supplementary table S3 and S9, Major circulating variant has been added in line 266-269 and table 3. The result is more specifically highlighted in the result section as well (line 301-304, 359-364 and 296-300). At the time of this study, variant of concerns was not classified and tools like Frejya that uses similar SNPs frequency-based algorithm and database containing all variant of concerns can actually predict circulation variant fractions (https://github.com/andersen-lab/Freyja) was not available.

 5 Several discrepancies between MiSeq and MinION are presented. This is very important, and the authors should discuss it in detail. The discrepancies has been highlighted both in result section line 292-293 Supplementary table S7.

 6 The authors should describe in detail the quality control of the NGS libraries and the final sequencing output The quality control has been added in line 190-193 and 202-206.

 7 The quantification of viral load in WW samples and the "persons infected" formula are not clearly presented and described. The formula was taken from the previous similar research paper; all the procedure was already described in those literature so was not mentioned earlier. The explanation has been added in line 135-137.

 

 8 And most importantly, I cannot draw any specific and major conclusions from the results and discussion presented.

 The discussion has been updated to add some clarity, we intended to demonstrate and highlight importance of genomic surveillance for disease like SARS-CoV-2. However, it is not feasible always for country like Nepal where fundings are not readily available, in such condition using low-cost sequencing to assess the situation and then if something important comes up then go for Miseq or clinical genomic surveillance of the community would be better option.

 

Reviewer 2 Napit et al. studied the genomic surveillance of SARS-CoV-2 in sewage from Kathmandu, Nepal. The authors aimed to establish a genomic-based environmental surveillance system by monitoring SARS-CoV-2 in sewage samples via PCR and WGS approaches. Although the experiments in the manuscript are well-organized and the manuscript itself is also written in a good way, this reviewer can see right away the shortages of this study. The following are listed some major issues for the authors' consideration.

 SNo Comment Response

 1 One of the main issues is the data quality of the sequencing. I looked into some of the raw reads the authors deposited on GitHub. Since those data are directly related to the conclusions, it will not stand if such bad quality data was used. The author should consider toning down some of the conclusions from Fig. 6. All the data was not of bad quality, few had low sequencing coverage, which is common in environmental samples as they do not have high viral load to that of clinical samples that has been acknowledged with all the table on coverage and depths in supplementary table and also in the manuscripts. The figure 6 (now figure 4) has also been replaced with the Supplementary figure S1. Those low coverage samples had failed to distinguish lineage to high resolution (table no 3 has been added) for clarity. A line has been added in discussion highlighting the same (line 370-371)

 2 Considering the environmental variants which might cause damage to lots of biological samples, the mutations the authors found has simply after the virus particles expose to sewage. It is essential to show some correlation to some clinical results to see if the mutations were presented in society.

 The comparison with clinical samples were in discussion section but result was updated to contain variant calling data from consensus sequences in result section and table no. 3. 

 

 3 This reviewer really appreciates this kind of excellent effort in studying SARS-CoV-2 in developing countries like Nepal. The authors can discuss the significance of this study extensively since the novelty of this study is not apparent, especially in the methodology and results. One possible direction is to mention that sewage might become one of the possible infection routes for SARS-CoV-2, and we should be cautious since some mutations were found in environmental samples, such as sewage, in this study.

 The discussion (line 400-409) has been updated to highlight the effort needed to track pandemic in country like Nepal cheaply and quickly in time.

---

## [Decision Letter · Decision Letter 2]

14 Mar 2023

Rapid genomic surveillance of SARS-CoV-2 in a dense urban community using environmental (sewage) samples

PONE-D-22-13706R2

Dear Dr. Karmacharya,

We’re pleased to inform you that your manuscript has been judged scientifically suitable for publication and will be formally accepted for publication once it meets all outstanding technical requirements. However, I suggest to re-phrase the title as follows: Rapid genomic surveillance of SARS-CoV-2 in a dense urban community of Kathmandu valley using sewage samples.

Kind regards,

Basant Giri, Ph.D.

Academic Editor

PLOS ONE

Additional Editor Comments (optional):

Reviewers' comments:

Reviewer's Responses to Questions

**Comments to the Author**

1. If the authors have adequately addressed your comments raised in a previous round of review and you feel that this manuscript is now acceptable for publication, you may indicate that here to bypass the “Comments to the Author” section, enter your conflict of interest statement in the “Confidential to Editor” section, and submit your "Accept" recommendation.

Reviewer #2: All comments have been addressed

2. Is the manuscript technically sound, and do the data support the conclusions?

Reviewer #2: Yes

3. Has the statistical analysis been performed appropriately and rigorously? 

Reviewer #2: N/A

4. Have the authors made all data underlying the findings in their manuscript fully available?

Reviewer #2: Yes

5. Is the manuscript presented in an intelligible fashion and written in standard English?

Reviewer #2: Yes

6. Review Comments to the Author

Reviewer #2: I have no further comments on this revised manuscript and the updated version should be acceptable for Plos ONE.

7. PLOS authors have the option to publish the peer review history of their article (what does this mean?). If published, this will include your full peer review and any attached files.

Reviewer #2: No

---

## [Editor Report · Acceptance letter]

21 Mar 2023

PONE-D-22-13706R2 

Rapid genomic surveillance of SARS-CoV-2 in a dense urban community of Kathmandu Valley using sewage samples 

Dear Dr. Karmacharya:

I'm pleased to inform you that your manuscript has been deemed suitable for publication in PLOS ONE. Congratulations! Your manuscript is now with our production department. 

Kind regards, 

on behalf of

Dr. Basant Giri 

Academic Editor

PLOS ONE